# *mdka* Expression Is Associated with Quiescent Neural Stem Cells during Constitutive and Reactive Neurogenesis in the Adult Zebrafish Telencephalon

**DOI:** 10.3390/brainsci12020284

**Published:** 2022-02-18

**Authors:** Luisa Lübke, Gaoqun Zhang, Uwe Strähle, Sepand Rastegar

**Affiliations:** 1Institute of Biological and Chemical Systems-Biological Information Processing (IBCS-BIP), Karlsruhe Institute of Technology (KIT), Postfach 3640, 76021 Karlsruhe, Germany; luisa.luebke@kit.edu (L.L.); zhangg@ie-freiburg.mpg.de (G.Z.); uwe.straehle@kit.edu (U.S.); 2Centre of Organismal Studies, University Heidelberg, Im Neuenheimer Feld 230, 69120 Heidelberg, Germany

**Keywords:** adult neurogenesis, regeneration, quiescence, telencephalon, radial glial cell, neural stem cell, *mdka*, zebrafish

## Abstract

In contrast to mammals, adult zebrafish display an extraordinary capacity to heal injuries and repair damage in the central nervous system. Pivotal for the regenerative capacity of the zebrafish brain at adult stages is the precise control of neural stem cell (NSC) behavior and the maintenance of the stem cell pool. The gene *mdka*, a member of a small family of heparin binding growth factors, was previously shown to be involved in regeneration in the zebrafish retina, heart, and fin. Here, we investigated the expression pattern of the gene *mdka* and its paralogue *mdkb* in the zebrafish adult telencephalon under constitutive and regenerative conditions. Our findings show that only *mdka* expression is specifically restricted to the telencephalic ventricle, a stem cell niche of the zebrafish telencephalon. In this brain region, *mdka* is particularly expressed in the quiescent stem cells. Interestingly, after brain injury, *mdka* expression remains restricted to the resting stem cell, which might suggest a role of *mdka* in regulating stem cell quiescence.

## 1. Introduction

Neurogenesis is the production of new neurons from neural stem cells (NSCs) which will proliferate, differentiate into neurons, and consequently integrate into existing neuronal networks [1,2,3]. For more than a century, the dogmatic view has been held that neurogenesis in mammals can only be facilitated during embryonic stages, when the nervous system is forming [4]. Today, accumulating evidence is being discovered showing that neurogenesis can also be maintained at adult stages in the mammalian brain [4,5,6,7,8]. The genesis of new neurons at adult stages is called adult neurogenesis and provides the ability to facilitate growth, learning, or the repair of neuronal loss due to injury or disease [2,3]. The zebrafish, as part of the teleost family, maintains neurogenesis at adult stages because it is continuously growing and therefore needs to form new neurons abundantly [9,10,11,12,13]. Adult neurogenesis in the zebrafish occurs in different brain regions where NSCs are organized in distinct stem cell niches [2,3]. Zebrafish have a high abundancy of stem cell niches distributed throughout all the brain and display a high ability to regenerate neuronal loss without obvious deficits [2,11,12,13]. In mammals, the production of new neurons is limited to only a few stem cell niches, mostly located in the mammalian forebrain [14,15]. In addition, the formation of a glial scar in the mammalian brain leads to insufficient replacement of neurons by stem cells and therefore to reduced regenerative potential [3,16,17,18,19]. In order to battle neurodegenerative diseases such as Parkinson’s and Alzheimer’s, which are becoming increasing problems in modern society, deciphering the regulatory and molecular background of neural regeneration in zebrafish holds the promise of developing potential cures.

Generally, the process of neural regeneration is based on a complex network of transcription factors and regulators, signaling pathways, as well as non-coding areas of the genome, giving rise to non-coding RNA molecules [20,21,22]. It is pivotal that there is tight spatial and temporal control of this network to maintain the balance between homeostasis and regeneration [23,24]. One chance to battle neurodegenerative diseases is to identify key players of this complex network and reactivate neurogenesis from NSCs in the adult mammalian brain. For this challenge, the zebrafish is an ideal model organism [25,26,27].

In the zebrafish adult brain, one of the most extensively studied stem cell niches is located along the ventricular layer of the telencephalon (forebrain) of the zebrafish and is known as the ventricular zone [28,29]. This niche is densely populated by the cell bodies of radial glia cells (RGCs), which are considered the stem cells of the telencephalon [28,29,30]. RGCs can be divided into three main subtypes: Type I cells are quiescent cells which do not proliferate and show radial glial characteristics such as a triangular-shaped soma and long processes spanning the parenchyma [31,32]. Type II cells are activated and therefore actively proliferating [23,31]. They also still show glial characteristics and express the radial glia markers S100β and GFAP, like the type I cells, but type II cells additionally express the proliferation marker PCNA (proliferating cell nuclear antigen) [23,28,31,33]. Type II cells can divide, either symmetrically or asymmetrically, to give rise to two new type II cells or one type II and one type III cell, respectively [34]. Type III cells are neuroblasts which act as precursors for mature neurons [1,28,34]. They do not display glial characteristics anymore and are actively proliferating [28,31].

In an effort to decipher the molecular mechanisms underlying neural regeneration, we conducted a screen for genes which are differentially expressed after injury of the zebrafish telencephalon. Via RNA sequencing of telencephalic tissue, we identified hundreds of candidates which showed changes in expression in the lesioned compared to unlesioned telencephalic tissue [21,35]. One of those genes was *midkine a* (*mdka*), the zebrafish homologue of the human midkine (MK) gene [21,36,37]. MK is a heparin binding growth factor that plays an important role during embryonic development, especially in the development of the nervous system [36]. Furthermore, MK belongs to a small gene family containing only one other member: pleiotrophin (PTN) [36]. This family is conserved among vertebrates, but in the zebrafish, it additionally contains the *mdka* paralogue *midkine b* (*mdkb*), which arose due to the genome duplication during teleost evolution [36,37,38,39,40]. During early zebrafish development, the two paralogues differ in their expression and biological function: *mdka* is expressed in the neural tube and is responsible for the development of ventral floor plate cells, while *mdkb* is mainly expressed in the dorsal neural tube and regulates the formation of neural crest cells [37,41,42]. *mdkb* has also been shown to play a pivotal role during the formation of the neural plate border in zebrafish [42]. Recent reports also show that the constitutive expression of *mdka* and *mdkb* in the zebrafish retina differs but their expression patterns align during retinal regeneration [43].

*mdka* is a prime candidate to be investigated in the context of regulation of stem cell behavior during regeneration for several reasons: (1) orthologues of *mdka*, such as mk in salamander and the mammalian orthologue MK, have previously been shown to be important factors during regeneration in other vertebrates and tissues, e.g., limb regeneration in the Axolotl or muscle and liver regeneration in mice [44,45,46]. (2) MK was also found to be expressed in parts of the brain during rat development and later by neural precursor cells in mice [38,47]. (3) *mdka* expression is significantly upregulated during regeneration of different organs in the zebrafish, such as the retina, the fin, and the heart [43,48,49,50]. (4) Members of the midkine family, such as mk in salamander and *mdka* in the zebrafish, were shown to play a significant role during inflammation, which is a key process during regeneration of the nervous system in vertebrates [44,48,51,52,53]. However, this gene has not yet been investigated regarding its role during regeneration of the telencephalon in any organism.

Here, we investigate the expression of *mdk* genes in the adult zebrafish brain. We identify *mdka* as being exclusively expressed in quiescent RGCs (qRGCs), where its expression is also highly colocalized with *id1*, a marker for qRGCs [35,54]. We show that *mdka* is responding to injury by an upregulation of expression which is delayed and peaks at 5 dpl (days post-lesion). This change in expression cannot be observed for *mdkb*. Additionally, after injury the *mdka* expression remains restricted to qRGCs, indicating that *mdka* is expressed in type I stem cells during constitutive as well as during regenerative neurogenesis. This is an intriguing indication that *mdka* might be involved in maintaining the NSC pool by keeping the stem cells in a quiescent state.

## 2. Results

### 2.1. The Expression Patterns of mdka and mdkb in the Adult Zebrafish Brain Are Overlapping but Different

In the zebrafish genome, the gene *mdka* has a closely related paralogue, which is *mdkb* [38,39]. Accordingly, our first aim was to investigate and compare the expression patterns of *mdka* and its paralogue *mdkb* in the adult zebrafish brain to answer the question of whether these genes are involved in neurogenesis. We performed chromogenic in situ hybridization (ISH) on cross-sections of the whole brain using probes against *mdka* and *mdkb* (Figure 1). In the telencephalon, expression of both genes was detected along the ventricle of the dorsomedial and dorsolateral telencephalon (Dl and Dm, Figure 1A,D), and along the ventricular layer of the ventral and dorsal nuclei of the ventral telencephalon. (Vv and Vd, Figure 1A,D). *mdkb* expression was additionally visible in cells outside of the ventricular layer in the dorsal telencephalon (Figure 1D), as well as inside the ventral and dorsal nuclei of the ventral telencephalon (Vv and Vd, Figure 1D). 

In the diencephalon, *mdka* expression was observed in the periventricular grey zone (PGZ) of the optic tectum (TeO) and the lateral division of the valvula cerebelli (Val, Figure 1B). Additional expression could be detected in the preglomerular nucleus and the caudal zone of the periventricular hypothalamus (PG and Hc, Figure 1B). In the dorsal zone of the periventricular hypothalamus (Hd), cells surrounding the lateral recess of the diencephalic ventricle exhibited strong *mdka* staining as well (LR, Figure 1B). *mdkb* expression was also detected in the PGZ and in the LR (Figure 1E).

In the posterior part of the diencephalon, *mdka* expression was perceived at the periphery of the torus longitudinalis (TL, Figure 1C), while *mdkb* expression was additionally detectable in the torus lateralis (TLa, Figure 1F).

We decided to focus our further investigation on *mdka* instead of *mdkb* for two reasons: *mdka* expression is more restricted to the ventricular zone and *mdka* is upregulated in response to injury in RNA sequencing of bulk RNA from telencephalic tissue [21]. This upregulation could not be observed for *mdkb* [21].

### 2.2. mdka Is Expressed in RGCs in the Telencephalon

In order to gain a deeper understanding of the role *mdka* plays during regeneration, we next asked in which cell type *mdka* is expressed. To address this question, we combined fluorescent in situ hybridization (FISH) against *mdka* mRNA with Immunofluorescence (IF), using antibodies against S100β on cross-sections of the zebrafish telencephalon (Figure 2). Again, expression of *mdka* mRNA was detected along the ventricular zone of the telencephalon (Figure 2A–A‴,B–B‴) and mainly expressed in S100β + RGCs (white arrows, Figure 2A–A‴,B–B‴, quantification Figure 2F). However, we observed a lack of *mdka* expression between the ventral and dorsal nuclei of the ventral telencephalon (white arrowheads, Figure 2A–A‴). This region corresponds to the rostral migratory stream (RMS), which is an area with high proliferative potential, as it is occupied by committed progenitor cells called neuroblasts, and lacks RGCs [31]. To further confirm that *mdka* is not expressed in proliferating cells, we co-stained telencephalic cross-sections with the proliferation marker PCNA (Figure 2A,A″,A‴,B″–B‴). *mdka* expression was not detected in PCNA+ cells (yellow arrows Figure 2B,B″,B‴, quantification Figure 2F), demonstrating that *mdka* is a gene which is expressed in qRGCs. Additionally, we used an anti-HuC/D (Hu) antibody to label post-mitotic neurons and found no expression of *mdka* mRNA in the HuC/D+ neurons (arrows Figure 2D–D‴,E). When evaluating single-cell data from Lange et al. (2020), we detected a high correlation between *mdka* and the gene *id1*, which we previously identified as being exclusively expressed in qRGCs [35,54]. In order to confirm the co-localization between *id1* and the earlier observation that *mdka* expression is mainly located to qRGCs, we used the transgenic zebrafish line *Tg(id1-CRM2:gfp)*, which recapitulates the expression of the endogenous *id1* gene and marks qRGCs [54]. Indeed, co-expression of *mdka* mRNA with the signal for GFP was noted in 72.7% of cells (Figure 2G and white arrows Figure 2H–H‴, quantification Figure 2I), locating *mdka* expression to qRGCs. Taken together, the observed—and, by single-cell sequencing data, confirmed—co-localization with *id1* verifies that *mdka* is expressed in qRGCs.

### 2.3. mdka Expression Is Upregulated after Injury of the Adult Telencephalon

In order to model brain injury and trigger regeneration, we utilized a stab-wound assay during which we inserted a syringe needle through the skull into one hemisphere of the telencephalon, leaving the other hemisphere as uninjured control [55,56]. In an RNA sequencing experiment that was previously carried out in our lab, comparing expression levels of different genes between injured and uninjured telencephalic hemispheres at 5 days after the injury, *mdka* was found to show a 1.37-fold expression increase after injury (*p*-value = 2.68^−9^) [21]. In an effort to confirm these results, we carried out a chromogenic ISH for *mdka* mRNA on telencephalic cross-sections of adult brains at 5 dpl (Figure 3D). We found that indeed a higher expression of *mdka* mRNA could be observed in the injured left hemisphere (black arrows, Figure 3D), confirming the RNA sequencing results. To evaluate possible expression changes over time as a response to injury, as well as identifying a possible peak of *mdka* expression, we additionally conducted ISH experiments at different time points after the injury. Immediately after the injury, at 1 dpl, we could observe no difference in the expression of *mdka* between the injured and uninjured hemispheres (Figure 3B). In comparison to an uninjured telencephalon, the expression does not change (compare Figure 3A with Figure 3B). After 5 dpl the expression in the injured hemisphere decreases until it reaches baseline levels at 10 dpl (arrows Figure 3D,E). We also performed a quantitative reverse transcription PCR (RT-qPCR) experiment comparing the level of *mdka* expression in injured and uninjured hemispheres at the different time points, confirming the expression time course and determining a peak of expression between 3 dpl and 5 dpl (Figure 3F).

### 2.4. mdka Expression Remains Restricted to qRGCs after Injury

We next asked whether there would be a change in the cellular expression pattern of *mdka* after injury, as we found that, under constitutive conditions, expression is restricted to qRGCs. Additionally, studies in other zebrafish tissues suggest that *mdka* expression is associated with proliferating stem cells after lesion [43]. To this end, we again employed a combination of fluorescent ISH and IF using antibodies against S100β and PCNA on telencephalic cross-sections of injured brains at 5 dpl (Figure 4). The upregulation in expression was especially visible in the ventricular layer of the ventral telencephalic area (Figure 4A). After the injury, expression of *mdka* was still highly detectable in cells which are positive for the radial glia marker S100β and negative for the proliferation marker PCNA (arrows Figure 4B–B‴ and quantification Figure 4C). Additionally, we again observed a high co-expression of *mdka* mRNA with the signal for GFP in the *id1-CRM2:gfp* transgenic line (76.6% of cells), showing that the cellular location of *mdka* expression does not change and is still restricted to qRGCs (white arrows Figure 4E–E‴ and quantification Figure 4F) after an injury.

Taken together, the results indicate that *mdka* is expressed in a similar pattern as *id1*, not only with regard to the cellular location but also onset of expression after injury. Therefore, it might, like *id1*, be necessary to confer quiescence to NSCs in order to maintain the integrity of the stem cell pool.

## 3. Discussion

### 3.1. The Expression Patterns of mdka and mdkb Differ in the Telencephalon but Overlap in the Posterior Brain

This is the first study that reports the expression of *mdka* and its paralogue *mdkb* in the zebrafish adult brain, focusing the analysis on the forebrain. Briefly, expression of *mdka* and *mdkb* was detected along the ventricle of the ventral and dorsal telencephalon. This area is considered to be the stem cell niche of the telencephalon, an area where the cell bodies of RGCs, the stem cells of the telencephalon, reside [28]. Considering that both genes are highly expressed in this area, it is tempting to speculate that both are involved with controlling and regulating the behavior of NSCs. However, the expression pattern of *mdkb* was broader and spread out into the dorsal telencephalon and into the different nuclei of the ventral telencephalon. Judging from the expression pattern, it is likely that *mdkb* is also expressed in newborn neurons, leaving the stem cell area to migrate to their target tissue. 

In the diencephalon, mainly cells in the periventricular grey zone of the optic tectum and parts of the hypothalamus showed expression of *mdka* and *mdkb*. When scanning further through the brain, both genes were shown to be highly expressed at the edge of the torus longitudinalis. Furthermore, when comparing both expression patterns throughout the brain, it becomes obvious that the patterns are highly overlapping but are still distinct, which is in line with results that we obtained in an embryo ISH (data not shown) and data obtained from other tissues such as the zebrafish retina [43,57].

### 3.2. mdka Is Highly Expressed in Quiescent Type I Stem Cells in the Telencephalon during Constitutive and Regenerative Neurogenesis

In this work, we also observed that the expression of *mdka* is not detectable in mature neurons but co-localizes with S100β, a marker for RGCs, the NSCs of the telencephalon, which is in accordance with recently published single-cell data [58]. This result is also in line with previously reported data in other tissues where *mdka* expression is also located in the tissue-specific stem cells, such as the Müller glia of the adult zebrafish retina [43,59]. Additionally, in this study we observed that *mdka* expression is mainly restricted to cells which are negative for the proliferation marker PCNA under constitutive and regenerative conditions, leading to the conclusion that the gene is expressed by cells which are non-proliferating RGCs and therefore quiescent type I stem cells. This is also confirmed by co-staining with the transgenic line for *id1* which marks qRGCs [54]. In sharp contrast to that, several reports show that *mdka* is expressed in activated stem cells in different tissues, including the zebrafish retina, where *mdka* is expressed in proliferating Müller glia during regeneration [43]. Additionally, loss of *mdka* is associated with decreased proliferation after injury of the fin or after photoreceptor ablation in the retina [50,59]. This obvious contradiction is possibly due to the fact that, in the brain, the source of stem cells greatly differs from other tissues. In the zebrafish retina, for example, regeneration is dependent on Müller glia which first need to de-differentiate in order to re-enter the cell-cycle and generate multipotent progenitors [60]. Furthermore, in the zebrafish fin, de-differentiated osteoblasts provide progenitor cells to repair the fin rays after injury [61]. In contrast, in the telencephalon, there is a constitutive pool of quiescent neural stem cells which can be recruited upon injury [2]. *mdka* does not necessarily have to play the same role in the regenerative processes of these different tissues. An in-depth analysis of the regulation and function of *mdka* is needed to elaborate further on the specific role of *mdka* during brain regeneration in the adult zebrafish.

### 3.3. In Response to Brain Injury, mdka Expression Is Upregulated in a Delayed Fashion and Possibly Involved with Stem Cell Quiescence

As a response to a brain injury, several different processes take place in the brain and different factors are secreted. During an RNA sequencing experiment in our lab, *mdka* was picked up as one of the genes which showed a significant upregulation in the injured tissue compared to uninjured tissue at 5 dpl [21]. This is also one of the reasons why we focused on *mdka*: *mdkb* was not differentially expressed in response to the injury [21]. A change in expression in the stem cell area of the telencephalon points to a possible involvement of this gene with reactive neurogenesis. We could confirm the detected upregulation in an ISH experiment, as well as in a RT-qPCR experiment. However, when looking at the time course of the deregulation by observing different time points after injury, we detected that there is no upregulation of the *mdka* expression at 1 dpl. In fact, the peak of expression seems to be between 3 and 5 days after the injury. This indicates that the response of *mdka* to the injury is not an instantaneous one and therefore this gene is likely not involved with responses that are happening immediately after an injury, such as the inflammatory response [53,62]. Rather, it seems that *mdka* is involved with mechanisms which are setting off in a delayed fashion. This is an important mechanism to avoid depletion of the stem cell pool in the brain. The fact that even after the injury *mdka* is still expressed in qRGCs, and the observed co-expression with the gene *id1*, which reportedly is an important factor in keeping stem cell quiescence, substantiate this hypothesis [54,63]. Intriguingly, in organs other than the brain, Mdka function has already been investigated. It is, for example, known to regulate cell-cycle progression in the zebrafish retina through the downstream effector *id2a* [59,64]. The relationship between cell-cycle kinetics and the control of stem cell proliferation in the telencephalon by *mdka* was investigated firstly in our lab by blocking Mdka through a small molecule compound (iMDK) [65]. Secondly, we compared the number of proliferative cells in telencephalic cross sections of *mdka* mutants (*mdka^mi5001^*) [59] with sections of WT fish. Nevertheless, in both cases we could not detect a significant change in the proliferative behavior of NSCs (unpublished observations). This might be due to the compensatory mechanism of either the *mdka* paralogue *mdkb* or other signaling pathways, as it is clear that a complex process such as neurogenesis cannot solely rely on one signaling pathway. Upstream of *mdka*, a possible regulator of *mdka* could be the BMP pathway, since *id1* and *mdka* are highly co-expressed and *id1* is known to be a BMP responsive gene [54,63]. However, after blocking the BMP pathway with a small molecule inhibitor, under homeostatic neurogenic conditions, we could not observe a change in *mdka* expression. The same was true for inhibition experiments conducted for other major signaling pathways which are known to control stem cell activity such as Notch, Wnt, and FGF (unpublished observations) [66,67,68,69]. Possibly, *mdka* is another pivotal factor controlling stem cell behavior in the brain and sustaining the balance between quiescent and proliferating stem cells, but the detailed mechanism of action, as well as *mdka* regulation, need to be further investigated. 

## 4. Conclusions

In this study, we described *mdka* and *mdkb* expression in the telencephalon, as well as the expression pattern for the gene *mdka* at cellular resolution in response to stab injury of the adult telencephalon. Interestingly, *mdka* is upregulated in response to injury and it is mostly expressed in qRGCs of the telencephalon, suggesting a key role of this gene in the control of stem cell behavior under constitutive and regenerative conditions. Our studies add *mdka* to the molecular signature of qRGCs. However, in order to elaborate more on the role of *mdka* during adult brain regeneration and confirm a possible involvement with the maintenance of the stem cell pool in the telencephalon, detailed studies on the function and regulation of *mdka* are needed. 

## 5. Material and Methods

### 5.1. Zebrafish Strains and Husbandry

Experiments were performed on 8–12-month-old AB wild-type (WT), *Tg(id1-CRM2:gfp)* [54] zebrafish. Zebrafish housing and husbandry were performed following the recommendations by [70]. All animal experiments were carried out in accordance with the German protection standards and were approved by the Government of Baden-Württemberg, Regierungspräsidium Karlsruhe, Germany (Aktenzeichen 35.9185.81/G288/18, G215/21).

### 5.2. Stab Wound

Stab-wound experiments on adult zebrafish were performed as described in [55,56]. After anesthetizing the fish with Tricaine, a syringe needle was inserted through the skull into the left hemisphere, while the contralateral right hemisphere was kept unlesioned and served as the control. 

### 5.3. Constructs and Synthesis of Antisense RNA DIG Probes

The following antisense digoxigenin-labeled probes were used: *mdka* and *mdkb*. The pcs2+ plasmids containing either *mdka* or *mdkb* cDNA were kindly provided by Christoph Winkler, National University of Singapore. For probe synthesis, 1 µg of the plasmid was linearized for 30 min at 37 °C using the appropriate restriction enzyme (see Table 1). After deactivation of the restriction enzyme for 5 min at 80 °C, the linearized plasmid was used for in vitro transcription with the DIG labeling mix (Roche; Basel, Switzerland) and RNA polymerase (see Table 1). The mixture was incubated for 3 h at 37 °C. Afterwards, the reaction was stopped by adding 0.2 M EDTA (pH8) and purified using the ProbeQuant G50 Micro column kit (GE Healthcare, Chicago, IL, USA). The probe was then diluted 1:1 using hybridization buffer [55] and stored at −20 °C. 

### 5.4. Preparation of Adult Zebrafish Brains, In Situ Hybridization, Immunofluorescence and Imaging, and Image Analysis

Preparation of brains (dissection and sectioning) for ISH and IF was performed as described in [55]. For IF, primary antibodies included: chicken anti-GFP (1:1000, Aves labs, Davis, CA, USA), mouse anti-PCNA (1:500, Dako, Agilent, Santa Clara, CA, USA), rabbit anti-S100β (1:400, Dako, Agilent, Santa Clara, CA, USA), and rabbit anti-HuC/D (1:500, Abcam, Cambridge, UK). Secondary antibodies were conjugated with Alexa fluor dyes (Alexa series) and included anti-chicken Alexa 488, anti-mouse Alexa 546, and anti-rabbit Alexa 633. For ISH, the prepared DIG probes were hybridized with the whole brain tissue. After cutting, secondary DIG antibodies (anti-DIG-AP for chromogenic staining; anti-DIG-POD for fluorescent staining) were applied overnight. Staining was conducted with NBT/BCIP (Perkin Elmer, Waltham, MA, USA) in the case of chromogenic staining or Tyramide Cy3 solution (Perkin Elmer, Waltham, MA, USA) for fluorescent staining. Pictures of chromogenic in situ hybridized sections were acquired with a Leica stereomicroscope MZ16 F. Immunohistochemically stained brain slices were mounted using Aqua-Poly/Mount (Cat No. 18606-20, Polysciences, Inc, Warrington, PA, USA) with coverslips (0.17 mm thickness) and imaged with a laser scanning confocal microscope (Leica TCS SP5). To obtain single-cell resolution images, an HCX PL APO CS × 63/1.2NA objective was used with the pinhole size set to 1 airy unit. Fluorescent images for green (GFP), red (*mdka* mRNA, PCNA), and infrared channels (S100β and HUC/D) were acquired sequentially in 16-bit color depth with excitation/emission wavelength combinations of 488 nm/492–550 nm, 561 nm/565–605 nm, and 633 nm/650–740 nm, respectively. Pixel resolution for XY and Z planes were 0.24 and 0.50 μm, respectively. For individual brain samples, at least three transverse sections were cut with a vibratome (VT1000S, Leica) and different anterior-posterior levels representing anterior, posterior, and intermediate telencephalic regions were imaged. 

Confocal brain images were opened with Fiji/ImageJ software as composite hyperstacks to manually evaluate co-localization of GFP, PCNA, and S100β proteins and expression of the *mdka* mRNA (FISH). Cells expressing individual markers or marker combinations were counted in the dorsomedial and the dorsolateral ventricular zones in three transverse sections prepared at different anteroposterior levels of the telencephalon.

### 5.5. Real-Time Quantitative PCR

Total RNA was isolated from adult telencephala using Trizol (Life Technology, ThermoFisher Scientific, Darmstadt, Germany). First-strand cDNA was synthesized from 1 µg of total RNA with the Maxima First-Strand cDNA synthesis kit (ThermoFisher Scientific, Darmstadt, Germany) according to the manufacturer’s protocol. A StepOnePlus Real-Time PCR system (Applied Biosystems, ThermoFisher Scientific, Darmstadt, Germany) and SYBR Green fluorescent dye (Promega, Madison, WI, USA) were used. Expression levels were normalized using *β-actin*. The relative levels of mRNAs were calculated using the 2^−ΔΔCT^ method. The primer sequences are listed in Table 2. Experiments were performed with at least 3 technical replicates, each time with RNA pooled from 3 WT brains.

### 5.6. Statistical Analysis

For the quantification of proliferating NSCs, the number of S100β+/PCNA+ type II cells expressing *mdka* or the number of cells expressing the *Tg(id1-CRM2:gfp)* transgene and *mdka* was counted in 1 μm steps of 50 µm-thick z-stacks (imaged with a 63 × objective). Three sections per brain from at least three individuals were analyzed. Comparisons between two data sets, from quantification of proliferating NSCs or RT-qPCR, were performed by Welch two-sample *t*-test. 

## Figures and Tables

**Figure 1 brainsci-12-00284-f001:**
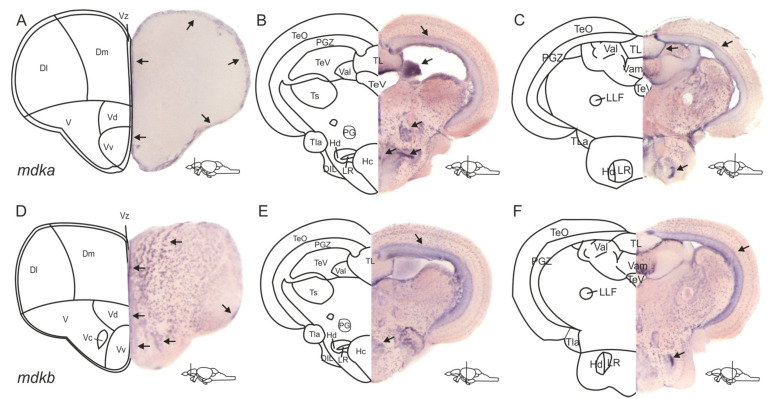
Expression patterns of the two paralogues *mdka* and *mdkb* differ in the adult zebrafish brain. (**A**–**F**) In situ hybridization (ISH) against *mdka* (**A**–**C**) and *mdkb* (**D**–**F**) on cross-sections through the zebrafish adult brain. *mdka* staining is visible along the ventricular layers of the dorsal and ventral telencephalic areas (**A**) and in the periventricular grey zone of the optic tectum and the lateral recess of the periventricular hypothalamus (**B**,**C**). In the diencephalon, *mdka* expression is additionally observed in the valvula cerebelli, the preglomerular nucleus, and the periventricular hypothalamus (**B**). *mdkb* expression is also detected in and outside of the ventricular layers of the telencephalon and spreads into the nuclei of the ventral telencephalon (**D**). In the diencephalon, it is also visible in the periventricular grey zone and the lateral recess (**E**,**F**). Abbreviations: D: dorsal telencephalic area; Dl: lateral zone of D; DIL: diffuse nuclei of the inferior lobe; Dm: medial zone of D; Hc: caudal zone of the periventricular hypothalamus; Hd: dorsal zone of the periventricular hypothalamus; LLF: lateral longitudinal fascicle; LR: lateral recess of the diencephalic ventricle; PG: preglomerular nucleus; PGZ: periventricular grey zone of TeO; TeO: optic tectum; TeV: telencephalic ventricle; TL: torus longitudinalis; TLa: torus lateralis; Ts: torus semicularis; V: ventral telencephalic area; Val: lateral division of the valvula cerebelli; Vam: medial division of the valvula cerebelli; Vc: central nucleus of V; Vd: dorsal nucleus of V; Vv: ventral nucleus of V; Vz: ventricular zone of the telencephalon.

**Figure 2 brainsci-12-00284-f002:**
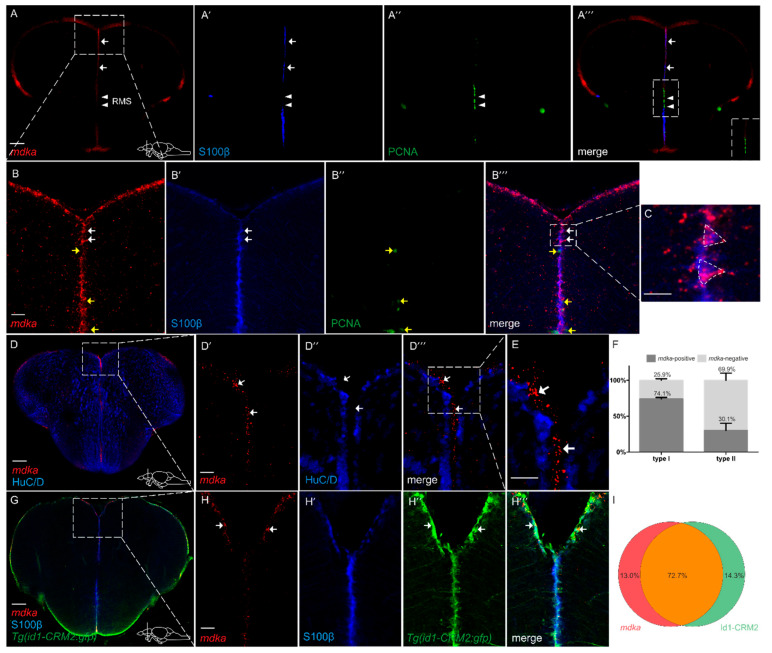
*mdka* is mainly expressed in non-proliferating RGCs in the telencephalon (**A**–**A‴**,**B**–**B‴**). Fluorescent in situ hybridization (FISH) with a probe directed against *mdka* mRNA (red) combined with Immunofluorescence (IF) with antibodies against PCNA (green) and S100β (blue) on cross-sections of WT telencephala. mRNA expression is detected in S100β+ cells and not detected in cells which are positive for PCNA. White arrows indicate *mdka*+/S100β+ cells. Yellow arrows indicate *mdka*−/PCNA+ cells. White arrowheads point to RMS. (**A‴**) inset: magnified view of the RMS, showing no co-expression between *mdka* mRNA and PCNA. Boxed-in area in (**A**) represents area of magnification in (**B**–**B‴**). Boxed-in area in (**B‴**) represents area of magnification in (**C**), showing two radial glial cells with their characteristic triangular shape and high expression of *mdka*. (**D**–**E**) FISH against *mdka* mRNA (red) with IF against the neuronal marker HuC/D (blue) indicating that *mdka* is not expressed in mature neurons since there is no co-localization between the two signals. Boxed-in area in (**D**) represents area of magnification in (**D′**–**D‴**). Boxed-in area in (**D‴**) represents area of magnification in (**E**). White arrowheads indicate *mdka*+/HuC/D− cells. (**F**) Quantification of S100β+, PCNA− type I and S100β+, PCNA+ type II cells expressing *mdka* mRNA. (**G**–**H‴**) FISH against *mdka* mRNA (red) combined with IF with antibodies against S100β (blue) on brains of the *Tg(id1-CRM2:gfp)* transgenic line (GFP, green). Expression of *mdka* mRNA is highly co-localized with cells positive for the transgene (green). White arrows show *mdka*+/*id1-CRM2*+ cells. Boxed-in area in (**G**) represents area of magnification in (**H**–**H‴**). (**I**) Quantification of cells expressing *mdka* mRNA in the *Tg(id1-CRM2:gfp)* transgenic line. Scale bar = 100 µm (**A**–**A‴**,**D**,**G**), 20 µm (**B**–**C**,**D′**–**E**,**H**–**H‴**). Location of cross-sections is indicated in lower right-hand corner of (**A**,**D**,**G**), respectively. *n* = 9 sections for quantifications. Abbreviations: *id1*, inhibitor of DNA binding 1; RMS, rostral migratory stream.

**Figure 3 brainsci-12-00284-f003:**
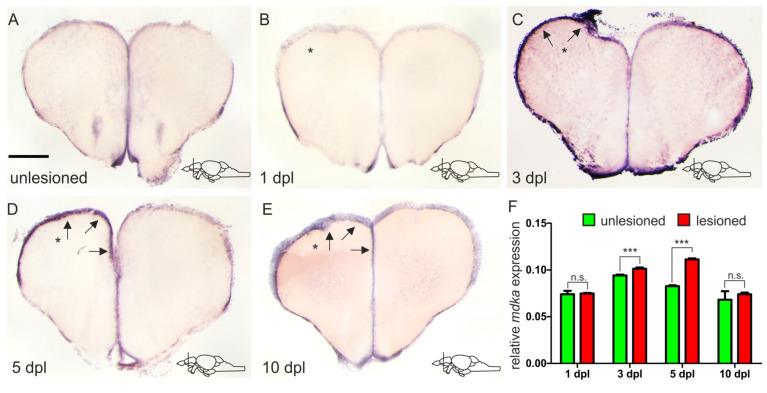
*mdka* is upregulated in a delayed fashion in response to injury. (**A**–**E**) ISH against *mdka* on cross-sections of the adult telencephalon in an unlesioned brain (**A**) and at different time points after the lesion (**B**–**E**). (**B**) Immediately after inflicting the lesion, there is no upregulation of *mdka* expression in the injured hemisphere. (**C**–**E**) *mdka* expression is increased in the injured hemisphere from 3 dpl onward (**C**) with a peak at 5 dpl (**D**), and then returns to baseline levels at 10 dpl (**E**). * The left, injured hemisphere is marked by an asterisk. Black arrows point at the expression of *mdka* in the ventricular zone of the left injured hemisphere. Location of cross-sections is indicated in the lower right-hand corner of (**A**–**E**). Scale bar = 100 µm. (**F**) RT-qPCR of *mdka* mRNA levels at different points after the lesion. Significance is indicated by asterisks: n.s. = not significant, *** *p* < 0.001. *n* = 3 brains/time point. Abbreviations: dpl, days post-lesion.

**Figure 4 brainsci-12-00284-f004:**
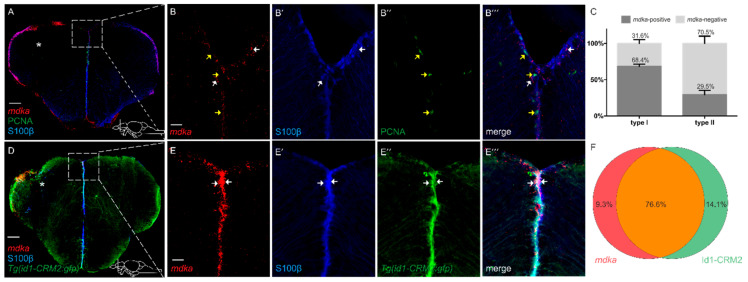
After injury, *mdka* is still highly expressed in S100β+/PCNA− type I cells. (**A**–**B‴**) FISH against *mdka* mRNA (red) combined with IF using antibodies against S100β (blue) and PCNA (green) on telencephalic cross-sections at 5 dpl. After injury, *mdka* mRNA is highly co-expressed with S100β and not detected in cells which are positive for PCNA. Boxed-in area in (**A**) represents area of magnification in (**B**–**B‴**). White arrows point to *mdka*+/S100β+ cells. Yellow arrows show *mdka*−/PCNA+ cells. (**C**) Quantification of S100β+, PCNA− type I and S100β+, PCNA+ type II cells expressing *mdka* mRNA after injury. (**D**–**E‴**) FISH against *mdka* mRNA (red) combined with IF using antibodies against S100β (blue) and GFP (green) on brains of the *Tg(id1-CRM2:gfp)* transgenic line at 5 dpl. The signal for *mdka* mRNA is strongly co-localized with cells positive for the transgene. Boxed-in area in (**D**) represents area of magnification in (**E**–**E‴**). White arrows indicate *mdka*+/S100β+/*id1*+ cells. (**F**) Quantification of cells expressing *mdka* mRNA in the *Tg(id1-CRM2:gfp)* transgenic line at 5 dpl. The injured hemisphere is indicated with an asterisk. Scale bar = 100 µm (**A**,**D**), 20 µm (**B**–**B‴**,**E**–**E‴**). Location of cross-sections is indicated in (**A**,**D**), respectively. *n* = 9 sections for quantifications.

**Table 1 brainsci-12-00284-t001:** Restriction enzymes and RNA-Polymerases used for DIG probe synthesis.

Gene Name	Restriction Enzyme	RNA-Polymerase
*mdka*	BamHI	T7
*mdkb*	BamHI	T7

**Table 2 brainsci-12-00284-t002:** Primer sequences for RT-qPCR.

*mdka* qPCR fw	5′-CGA CAC AGA AAA CAA AAT GCG GG-3′
*mdka* qPCR rev	5′-TAG AGC CAC TCC GCA CAG TC-3′
*β-actin* fw	5′-GCC TGA CGG ACA GGT CAT-3′
*β-actin* rev	5′-ACC GCA AGA TTC CAT ACC C-3′

## Data Availability

The sequencing data are available in the Gene Expression Omnibus database under accession number GSE137525. The source code for the computational analysis is available under github.com/fabianrost84/lange_single-cell_2019 (accessed on 14 December 2021). mRNAseq and small RNAseq data have been deposited in the Gene Expression Omnibus data base under the accession identifiers GSE161137 and GSE160992, respectively.

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
