# Peer review of "mdka Expression Is Associated with Quiescent Neural Stem Cells during Constitutive and Reactive Neurogenesis in the Adult Zebrafish Telencephalon"

_brainsci, 2022, doi:10.3390/brainsci12020284_

Round 1
Reviewer 1 Report
The authors presented their work very good about the expression pattern of the gene mdka and its paralogue mdkb in the zebrafish adult telencephalon under constitutive and regenerative conditions. mdka is particularly expressed in the quiescent stem cells. After brain injury mdka expression remains restricted to the resting stem cell, which might suggest a role of mdka in regulating stem cell quiescence.
Author Response
We are grateful to reviewer #1 for her/his time and effort to read and review our manuscript and for the positive comments.
Reviewer 2 Report
Thank you for allowing me to review this manuscript entitled: “mdka expression is associated with quiescent neural stem cells 2 during constitutive and reactive neurogenesis in the adult 3 zebrafish telencephalon”. The original article aimed to determine the expression of mdka and mdkb in the zebrafish telencephalon in the absence and presence of brain damage. The authors documented differential expression patterns of both molecules. Mdka had a more restricted expression to stem cell niches, its expression increases in response to damage. Mdkb expression is less specific and is not modified in response to damage. Mdka is expressed in qRGCs but not in neurons or proliferating cells. The authors conclude that mdka may contribute to repair after brain damage by participating in regulating the quiescent cell population.
The following aspects should be considered:
- The introduction and discussion are well-argued, but the references need to be updated.
- Figs 1 and 3 should include magnifications that demonstrate the expression in greater detail.
- The term immunohistochemistry is usually used for chromogenic stains that require an enzymatic reaction. The term Immunofluorescence is preferable for the methodology used.
- In vitro should be italics (line 356).
- Involvement in some pathways that illustrate the possible mechanisms of action of mdka should be added to the study.
- The conclusion is supported by its results.
Author Response
Reviewer: 2
Comments and Suggestions for Authors:
- The introduction and discussion are well argued, but the references need to be updated.
Thank you to reviewer #2 for this suggestion. We added the following references to the manuscript:
line 30:
[7] Altman et al., 1969
[8] Altman et al., 1965
line 32:
[2] Ghaddar et al., 2021
line 35:
[10] Pellegrini et al., 2005
[11] Zupanc et al., 2005
[12] Lindsey and Tropepe, 2006
[13] Edelmann et al., 2013
line 52:
[25] Kizil et al., 2012
[26] Grandel and Brand, 2013
[27] Schmidt et al., 2013
line 55:
[28] Pellegrini et al., 2007
[29] Lindsey et al., 2012
line 72
[37] Winkler et al., 2003
- Figs 1 and 3 should include magnifications that demonstrate the expression in greater detail.
We acknowledge the critique of reviewer #3 regarding the figures 1 and 3. However, chromogenic in situ hybridization, which was used in the demonstrated experiments, unfortunately does not allow cellular resolution. With these figures the purpose is solely to show an overview over the regions where mdka and mdkb are expressed (figure 1) or to permit a comparison between the mdka expression levels of injured and uninjured brain tissue (figure 3). In order to go into detail and investigate the cellular location of mdka expression, we employed the method of fluorescent in situ hybridization in combination with Immunofluorescence as it is demonstrated in figures 2 and 4. We added detailed magnifications on a few parts of the figures 2 and 4 to enable the observation of greater cellular detail.
- The term immunohistochemistry is usually used for chromogenic stains that require an enzymatic reaction. The term immunofluorescence is preferable for the methodology used.
With our sincere thanks we acknowledge this suggestion of reviewer #3. We have changed the text accordingly.
- in vitro should be italics (line 356)
We are grateful to reviewer #2 for pointing out this typo and have fixed it.
- Involvement in some pathways that illustrate the possible mechanisms of action of mdka should be added to the study.
We thank reviewer #2 for this suggestion. We added the following paragraph to the result section of our manuscript:
Intriguingly, in other organs than the brain, Mdka function has already been investigated. It is, for example, known to regulate cell-cycle progression in the zebrafish retina through the downstream effector id2a. The relationship between cell-cycle kinetics and the control of stem cell proliferation in the telencephalon by mdka was investigated in our lab by blocking Mdka through a small molecule compound (iMDK). However, this did not lead to a significant change in the proliferative behavior of NSCs (unpublished observation). Upstream of mdka, a possible regulator of mdka could be the BMP pathway, since id1 and mdka are highly co-expressed and id1 is known to be a BMP responsive gene. However, after blocking the BMP pathway with a small molecule inhibitor, under homeostatic neurogenic conditions, we could not observe a change in mdka expression. The same was true for inhibition experiments conducted for other major signaling pathways which are known to control stem cell activity like Notch, Wnt and FGF (unpublished observations).
Reviewer 3 Report
Section: Material and methods
Line 348: Skull instead of skill.
Author Response
Reviewer: 3
Comments and Suggestions for Authors:
- Section: Material and Methods, Line 348: Skull instead of skill
We thank reviewer #3 for pointing out this mistake. We have fixed the typo and proofread the manuscript to eliminate all spelling mistakes.
Reviewer 4 Report
The gene mdka, a member of a small family of heparin binding growth factors and they investigated the expression pattern of the gene mdka and its paralogue mdkb in the zebrafish adult telencephalon under constitutive and regenerative conditions. Their findings show that only mdka expression is specifically restricted to the telencephalic ventricle, a stem cell niche of the zebrafish telencephalon. In this brain region mdka is particularly expressed in the quiescent stem cells. Interestingly, after brain injury mdka expression remains restricted to the resting stem cell, it is suggested a role of mdka in regulating stem cell quiescence.
As the authors mention in the Introduction, MK plays an important role during embryonic development, especially in the development of the nervous system in mammals and they finding of the expression of Mdka in the zebrafish adult telencephalon (quiescent neural stem cells) may be important against constitutive and reactive neurogenesis as in brain injury. To further confirm the function of this factor (not only the expression pattern), I recommend to use Mdka mutant which looks like it is available to further support their hypothesis (eg. Population of RGC (Type1-3) can be changed in the mutant? or the neurogenesis state after injury can be changed in the mutant?etc) for improving their manuscript.
Author Response
Reviewer: 4
Comments and Suggestions for Authors:
- As the authors mention in the Introduction, MK plays an important role dung embryonic development, especially in the development of the nervous system in mammals and they finding of the expression of Mdka in the zebrafish adult telencephalon (quiescent neural stem cells) may be important against constitutive and reactive neurogenesis as in brain injury. To further conform the function of this factor (not only the expression pattern), I recommend to use Mdka mutant which looks like it is available to further support their hypothesis (eg. Population of RGC (Type1-3) can be changed in the mutant? Or the neurogenesis state after injury can be changed in the mutant?etc) for improving their manuscript.
We are very thankful to reviewer #4 for raising this important and very interesting point. We are aware of the existence of a mdka mutant line (mdkami 5001) in the lab of Prof. Peter Hitchcock at the Kellogg Eye Center in Michigan and agree with the reviewer that the investigation of this mutant line would have been a nice addition to our study. Unfortunately, the procedure of shipping fish overseas is quite complicated and due to the pandemic even more difficult. Additionally, the process of getting fish through quarantine in our facility, raising them and identifying homozygous mutants is fairly time consuming and would have gone beyond the time scope of this paper. However, we contacted Peter Hitchcock and he and his lab were so kind to check a possible change in proliferation in the adult telencephalon of the mutant fish compared to wt siblings by Immunostaining of cryo sections. Unfortunately, we could not detect a considerable difference in the proliferative behavior of stem cells. This might be explained by a redundant function of the mdka paralogue mdkb in the zebrafish. Furthermore, several independent pathways and molecules control neural stem cell behavior which might also be an explanation why there was no noticeable difference as they could be compensating for the loss of one molecule.
Round 2
Reviewer 4 Report
The authors have performed an experiment to prove the function of mdka in stem cell proliferation and found no considerable difference in WT vs MT possibly due to a redundant function of the mdka paralogue mdkb in the zebrafish. I think this result should be mentioned in the Discussion (or result) to clarify their research limitation and to provide an idea of what kind of experiments (eg double MTs) to be performed to further assess the role of mdka in quiescent neural stem cells of the adult zebrafish telencephalon.
Author Response
Changes in response to the reviewer#4:
Reviewer: 4
Comments and Suggestions for Authors:
The authors have performed an experiment to prove the function of mdka in stem cell proliferation and found no considerable difference in WT vs MT possibly due to a redundant function of the mdka paralogue mdkb in the zebrafish. I think this result should be mentioned in the Discussion (or result) to clarify their research limitation and to provide an idea of what kind of experiments (eg double MTs) to be performed to further assess the role of mdka in quiescent neural stem cells of the adult zebrafish telencephalon.
We are very thankful to reviewer #4 for raising this point.
we have added a sentence at the end of discussion section where we explain that we performed an experiment to prove the function of mdka in stem cell proliferation and found no considerable difference in the telencephalon of WT vs mdka mutants.